# Rare Bacteria Can Be Used as Ecological Indicators of Grassland Degradation

**DOI:** 10.3390/microorganisms11030754

**Published:** 2023-03-15

**Authors:** Mengjun Liu, Yue Ren, Weihong Zhang

**Affiliations:** 1Institute of Animal Husbandry and Veterinary Science, Tibet Academy of Agricultural and Animal Husbandry Sciences, Lhasa 850000, China; 2Key Laboratory of Aquatic Botany and Watershed Ecology, Wuhan Botanical Garden, Chinese Academy of Sciences, Wuhan 430074, China; 3University of Chinese Academy of Sciences, Beijing 100049, China

**Keywords:** grassland degradation, bacterial community, community assembly, co-occurrence patterns, Qinghai-Tibet Plateau

## Abstract

Grassland degradation is a major ecological problem at present, leading to changes in the grassland environment and the soil microbial community. Here, based on full-length 16S rRNA gene sequencing, we highlight the importance of small-scale environmental changes on the Qinghai-Tibet Plateau grassland for the composition and assembly processes of abundant and rare bacterial taxa. The results showed that grassland vegetation coverage affected the taxonomic and phylogenetic composition of rare bacterial taxa more than abundant bacterial taxa. The taxonomic composition and phylogenetic composition of rare bacterial taxa were also affected by soil nutrients. The relative contribution of deterministic processes (variable selection and homogeneous selection) to rare bacterial taxa was higher than that of the abundant bacterial taxa. The competitive potential within rare bacterial taxa was lower than that of the competitive potential between rare and non-rare bacterial taxa or within non-rare bacterial taxa. The assembly of rare bacterial taxa was more susceptible to environmental changes caused by grassland degradation than the abundant bacterial taxa. Furthermore, the distribution of rare bacterial taxa in the different degraded grassland soil was more local than that of abundant bacterial taxa. Thus, rare bacterial taxa could be considered an ecological indicator of grassland degradation. These findings help to improve our understanding of the composition and assembly mechanism of the bacterial communities in degraded grassland and provide a basis for the establishment of the grassland degradation management strategy.

## 1. Introduction

The grassland ecosystem covers more than 60% of the Qinghai-Tibet Plateau and provides important ecosystem functions, such as pastoral production, soil and water conservation, and biodiversity protection [1]. Grassland ecosystem health refers to the maintenance degree of land, vegetation, water, air, and ecological processes in the grassland ecosystem. Healthy grasslands maintain ecological balance and species diversity while providing continuous utilization opportunities for grassland livestock producers and supporting a range of production practices [2]. However, the grassland ecosystem may be more susceptible to environmental pressures than other terrestrial ecosystems [3], especially as the unique geographical environment and the climate of the Qinghai-Tibet Plateau make the grassland ecosystem more fragile [4]. Therefore, grassland on the Qinghai-Tibet Plateau is facing serious degradation problems [1,5]. Grassland degradation not only affects vegetation characteristics but also changes soil physicochemistry and nutrient characteristics, which directly affects the soil microbial community [6].

Microbial communities play a vital role in the soil biogeochemical cycles and in maintaining ecosystem sustainability [7]. Bacterial communities are a major component of biodiversity in the soil and play a vital ecological role in the global biogeochemical cycles [8]. The heterogeneity of environmental stress adaptation and substrate preference led to the differences in the growth and biomass yield of the microbial community [9,10]. This results in an often skewed abundance distribution in local microbial communities, where a large number of rare species coexist with a relatively small number of dominant species [10,11]. Some studies have confirmed that rare and abundant bacterial taxa in different soil samples often show different distribution patterns and functional traits [10,12,13], especially as rare species are increasingly recognized as vital and yet still vulnerable components of the earth’s ecosystems [14]. Grassland degradation will also lead to changes in the soil environment, which will affect the composition of microorganisms. Therefore, unraveling the composition, biogeographical distribution, and assembly processes of abundant and rare microbial taxa in degraded grassland soils on the Qinghai-Tibet Plateau is critical to understanding microbial-driven ecosystem processes.

Previous studies on the effects of grassland degradation on soil nutrients and microbial communities on the Qinghai-Tibet Plateau have shown that grassland degradation can lead to changes in the composition of microbial communities [5,15]. These studies provide a good perspective for understanding the soil microbial composition of different degraded grassland. However, grazing will lead to different grassland degradation of the same grassland types under the same climatic environmental region. Little is known, in fact, about the composition and assembly of abundant and rare microbial taxa in degraded grassland soils and their driving mechanisms at small region scales.

Given the unique geographical and climatic conditions, clarifying the relationship of composition and the assembly of soil abundant and rare bacterial taxa with vegetation and soil characteristics on the Qinghai-Tibet Plateau is of great significance to understanding the ecological mechanism of soil bacterial communities under different degraded grasslands. Thus, this study takes the typical mountain meadow of the Qinghai-Tibet Plateau in Bailang County, Xigaze City, Tibet, China as the research object, highlighting the importance of small-scale environmental changes on the Qinghai-Tibet Plateau grassland for the composition and assembly processes of abundant and rare bacterial taxa via full-length 16S rRNA gene sequencing. We expect the results of this study to provide a basis for the health assessment and management of the grassland ecosystem on the Qinghai-Tibet Plateau by revealing the response of abundant and rare bacterial taxa to grassland degradation, and the identification of key soil environmental factors related to grassland degradation.

## 2. Methods and Materials

### 2.1. Sampling Site and Sampling

The sampling area (89°20′ E, 29°13′ N) is located in Bailang County, Xigaze City, Tibet, China, with an altitude of 4300 m, which is a typical mountain meadow of the Qinghai-Tibet Plateau (Figure 1). This region belongs to the plateau temperate semi-arid monsoon climate, with an average annual rainfall of 360 mm and an average annual temperature of 5.9 °C. The soil samples were collected in the grassland after grazing Tibetan sheep. A total of 18 sampling sites were randomly set, and the distance between each two sampling sites was about 200 m. 0.5 × 0.5 m^2^. Parallel quadrats were set up at each sampling site to measure the vegetation coverage of grassland. The vegetation coverage was 5.25% to 69.96% (mean = 29.36%). Detailed data on vegetation coverage were shown in Appendix A. Five 0–20 cm surface soil subsamples in the 0.5 × 0.5 m^2^ parallel quadrats were then collected by a stainless steel cylindrical ring knife. Five subsamples from each site were mixed evenly as soil samples for that site. Part of the soil sample was frozen in liquid nitrogen and stored at −80 °C for the determination of the bacterial community composition. The other part of the sample was dried naturally for the determination of soil physicochemistry properties.

### 2.2. Analysis of Soil Physicochemical Properties

Soil physicochemical properties were discussed because of their importance in influencing the composition and structure of soil bacterial communities in cropland and grassland soil ecosystems [12,16]. In this study, the following soil physicochemical and nutrient variables were quantified according to the method described by Qiu et al. [17]. Soil physicochemical variables included pH and soil water content (SWC). Soil nutrient variables included total nitrogen (TN), total organic carbon (TOC), total phosphorus (TP), hydrolyzable nitrogen (HN), and available phosphorus (AP). Detailed data on soil physicochemical and nutrient variables are shown in Appendix A.

### 2.3. DNA Extraction and Full-Length 16S rRNA Gene Sequencing

Total DNA was extracted from each sample using the DNeasy PowerSoil Kit (QIAGEN, Inc., Venlo, The Netherlands), following the manufacturer’s instructions. Specific primers 27F (5′-AGRGTTYGATYMTGGCTCAG-3′) and 1492R (5′-RGYTACCTTGTTACGACTT-3′) were designed to target full-length amplicon. [18]. Target sequences were amplified by PCR and its products were purified, quantified, and homogenized to generate the SMRT bell library [19]. Library QC was performed on the libraries and the qualified ones were processed for sequencing on the PacBio Sequel platform [20]. The sequences were identified by barcode sequences, which resulted in the fastq sequences file of each sample. Dada2 in QIIME2 was applied to denoise sequences, generating ASVs [21]. Taxonomic annotation of feature sequences was processed by a Bayesian classifier and blast using Silva.138 as a reference database [22]. According to the method described by Wemheuer et al. [23], we use the package “Tax4Fun2” in R to analyze the function and functional redundancy index (FRI) of abundant and rare bacterial taxa.

### 2.4. Statistical Analysis

According to previous studies [24,25], we defined ASVs at the regional level with average relative abundances >0.10% as “abundant,” those with average relative abundances <0.01% as “rare”, and those in between as “intermediate”. The composition similarity of the bacterial community at taxonomic and phylogenetic levels was estimated based on the ASVs level via the “vegan” package and the “picante” package, respectively [13]. The assembly processes of the bacterial community were applied via the “picante” package, according to methods described in the previous study [13,26,27]. The potential associations of the bacterial community composition and the assembly processes on grassland environmental factors were calculated by SPSS 19.0. The niche width of bacterial communities was evaluated via the “spaa” package [28]. Co-occurrence networks of bacterial communities were constructed based on a strong (|*r|*> 0.7) and significant (*p*-value < 0.001) Spearman correlation at the ASVs level. The networks were visualized via Gephi, and the topological properties of the network were also calculated via Gephi.

## 3. Results

### 3.1. Diversity and Composition of Abundant and Rare Bacterial Taxa

The Shannon index curve of the sequence number showed that all soil samples were deeply sequenced and could be used to analyze soil bacterial communities in degraded grasslands (Figure 2A). A total of 193 ASVs of bacterial communities in all degraded grassland soil samples were classified as abundant bacterial taxa, and 2216 ASVs were classified as rare bacterial taxa. The average relative abundance of abundant bacterial taxa (54.16%) was higher than that of rare bacterial taxa (8.72%) (Figure 2B). However, the Chao1 richness (Figure 2C) and Shannon diversity (Figure 2D) of abundant bacterial taxa were lower than those of rare ones. This also means that rare bacterial taxa with a large number of species in the soil bacterial community of degraded grassland exist in the form of low abundance.

Figure 2E showed that the abundant bacterial taxa at the phylum taxonomic level were mainly composed of Acidobacteriota (24.48%), Verrucomicrobiota (24.42%), Proteobacteria (18.26%), Bacteroidota (13.26%), and Gemmatimonadota (6.60%), while the main composition of rare bacterial taxa was Proteobacteria (14.42%), Patescibacteria (14.42%), Bacteroidota (11.29%), Planctomycetota (10.72%), and Acidobacteriota (10.71%). The abundance-occupancy relationship showed that soil rare bacterial taxa had a stronger positive correlation than abundant bacterial taxa (Figure 2F). Meanwhile, abundant bacterial taxa in the different degraded grassland soil were more widespread than rare ones. The flower diagram showed that 129 ASVs of the abundant bacterial taxa were present in all soil samples (Figure 2G), while the rare bacterial taxa were not detected in all soil samples (Figure 2H). The niche width of the abundant bacterial taxa was also more than the rare bacterial taxa (Appendix A). This also indicates that the abundant bacterial taxa in the different degraded grassland soil were more regional, while the rare bacterial taxa were more local. However, even the relative abundance of these shared abundant bacterial taxa varied from sample to sample (Figure 2I).

### 3.2. Responses of Bacterial Communities to Vegetation Characteristics and Soil Physicochemical and Nutrients

The taxonomic similarity and phylogenetic similarity of the abundant bacterial taxa were higher than that of the whole bacterial community, while the taxonomic similarity and phylogenetic similarity of the rare bacterial taxa were lower than that of the whole bacterial community (Figure 3A,B). This indicates that the rare bacterial taxa in the different degraded grassland soil had more variation taxonomically and phylogenetically than the abundant bacterial taxa. The taxonomic similarity of the abundant and rare bacterial taxa was significantly positively correlated with their corresponding phylogenetic similarity, and the correlation of rare bacterial taxa was stronger than that of abundant bacterial taxa (Figure 3C,D), indicating that these rich and rare bacterial taxa at the taxonomic and phylogenetic level had different sensitivities to environmental changes. Moreover, the correlation between the taxonomic similarity of the abundant bacterial taxa and the whole bacterial community was stronger than that between the rare bacterial taxa and the whole bacterial community (Figure 3E). Meanwhile, the correlation between the phylogenetic similarity of the abundant bacterial taxa and the whole bacterial community was stronger than that between the rare bacterial taxa and the whole bacterial community (Figure 3F). This suggests that abundant bacterial taxa play a more important role than rare bacterial taxa in the taxonomic and phylogenetic variation of the whole bacterial community.

We further evaluated the responses of bacterial communities to vegetation characteristics and soil physicochemical and nutrients (Table 1). The taxonomic similarity of abundant bacterial taxa was affected by pH, SWC, TN, TOC, TP, and HN, while the taxonomic similarity of rare bacterial taxa was affected by vegetation coverage, SWC, TN, TOC, TP, HN, and AP. Among them, TN was an important factor affecting the variation of the taxonomic composition of abundant and rare bacterial taxa. However, the phylogenetic similarity of abundant bacterial taxa was only affected by pH, while the phylogenetic similarity of rare bacterial taxa was only affected by pH, vegetation coverage, SWC, TN, TOC, TP, and HN.

### 3.3. Assembly Processes of Abundant and Rare Bacterial Taxa in Degraded Grassland Soil

Figure 4A showed that the dispersal limitation contributed more to the assembly of abundant bacterial taxa (88.24%) in degraded grassland soils than rare bacterial taxa (45.10%), while the variable selection, homogeneous selection, homogenizing dispersal, and undominated contributed more to the assembly of rare bacterial taxa (13.73%, 3.92%, 5.23%, and 32.03%, respectively) than abundant bacterial taxa (11.11%, 0.00%, 0.00%, and 0.65%). Although the ecological assembly processes of abundant (88.89%) and rare bacterial taxa (82.35%) in degraded grassland soils were mainly stochastic processes, the relative contribution of stochastic processes to the assembly of different bacterial taxa in degraded grassland soil was different (Figure 4B). The relative contribution of differentiation to the assembly of abundant bacterial taxa (99.35%) in degraded grassland soils was greater than that of rare bacterial taxa (58.82%).

We evaluated the relationship between the assembly processes of abundant and rare bacterial taxa in degraded grassland soils with their corresponding vegetation characteristics and soil physicochemical and nutrients (Appendix A). Pairwise comparison βNTI values of soil abundant bacterial taxa had a significant positive correlation with pH, TOC, TP, HN, and AP, while the βNTI values of soil rare bacterial taxa had a significant positive correlation with pH, vegetation coverage, SWC, TN, TOC, TP, and HN. This also suggests that the assembly of abundant and rare bacterial taxa was influenced by different environmental factors. The βNTI values of soil abundant bacterial taxa also had a stronger correlation with TP (r = 0.272), while the βNTI values of soil rare bacterial taxa had a stronger correlation with TN (r = 0.304). This indicates that TP may be an important driving factor in the assembly of abundant bacterial taxa in degraded grassland soils, while TP may be an important driving factor in the assembly of rare bacterial taxa.

### 3.4. Co-Occurrence Network of Abundant and Rare Bacterial Taxa in Degraded Grassland Soil

The co-occurrence network was constructed based on strong (r > 0.7) and significant (*p*-value < 0.01) Spearman’s correlation relationships to understand the potential connections among different bacterial taxa in the degraded grassland soils (Figure 5A). The co-occurrence network of different bacterial taxa in degraded grassland soils were the scale-free small-world networks (Appendix A). Figure 5A showed that the network consisted of 1527 nodes linked by 8051 edges, with a much larger number of positive correlations (62.13%) than negative correlations (37.87%). The percentage of negative correlations within rare bacterial taxa (3.70%) was lower than those within abundant (37.98%) and intermediate bacterial taxa (37.01%). Moreover, the percentage of negative correlations was higher between the abundant and rare bacterial taxa (48.33%) than within themselves. Although more members of rare bacterial taxa (16.76%) participated in the construction of a co-occurrence network than those of abundant bacterial taxa (12.12%), the potential connections within abundant bacterial taxa (366) were higher than those within rare bacterial taxa (108). We compared unique node-level topological features of different bacterial taxa (Figure 5B). The degree, betweenness centrality, and eigenvector centrality values of abundant bacterial taxa were significantly higher than those of intermediate and rare bacterial taxa. However, the closeness centrality values showed no significant differences among these three bacterial taxa.

### 3.5. Potential Ecological Function of Abundant and Rare Bacterial Taxa in Degraded Grassland Soils

The abundant bacterial taxa in degraded grassland soils had a strong potential ecological function of environmental adaptation, xenobiotics biodegradation and metabolism, lipid metabolism, and metabolism of other amino acids (Figure 6A). However, the potential ecological function of metabolism nucleotide metabolism, metabolism of terpenoids and polyketides, metabolism of cofactors and vitamins, glycan biosynthesis and metabolism, global and overview maps (e.g., carbon metabolism and fatty acid metabolism), and biosynthesis of other secondary metabolites of abundant bacterial taxa in degraded grassland soils were significantly weaker than those of rare bacterial taxa (Figure 6A). Moreover, the potential ecological function of nitrogen metabolism, sulfur metabolism, and methane metabolism of abundant bacterial taxa in degraded grassland soils were all significantly stronger than those of rare bacterial taxa, while the oxidative phosphorylation of rare bacterial taxa was stronger than that of abundant bacterial taxa (Figure 6B). The FRI of rare bacterial taxa (7773) in degraded grassland soils was higher than that of abundant bacterial taxa (588), indicating that the probability of potential ecological function loss of rare bacterial taxa after grassland degraded disturbance was lower than that of abundant taxa (Figure 6C).

## 4. Discussion

Grassland degradation is a serious global ecological problem [29]. In recent decades, the grassland across the Qinghai-Tibet Plateau in China has been degraded to varying degrees due to the joint influence of human activities and climate change [5,7,30]. A soil bacterial community in the grassland is an important indicator to reflect environmental changes and the ecosystem status during grassland degradation because its response to environmental disturbance is rapid and strong [24,31]. Abundant and rare bacterial taxa coexisting in grassland soils form complex interspecific interaction systems. However, the assembly process of these abundant and rare bacterial taxa in degraded grasslands remains unclear. Therefore, this study revealed the effects of environmental factors caused by grassland degradation on the composition, assembly, and function of rare and abundant bacterial taxa from the Qinghai-Tibet Plateau. The results showed that abundant and rare bacterial taxa had different and complex responses to environmental changes under grassland degradation.

A total of 129 ASVs of abundant bacterial taxa persisted in degraded grassland and occupied a significant portion of the abundant bacterial taxa (Figure 2G). This suggests that the abundant bacteria have broader environmental adaptability than rare bacteria (Figure 2G,H), which is consistent with the observation of Wan et al. [10] that abundant bacteria have a higher environmental threshold than rare bacteria in the wetland soil of the Qinghai-Tibet Plateau. This phenomenon may be because abundant bacterial taxa have the potential for easier nutrient utilization than rare bacterial taxa [10,32]. The results of functional prediction showed that the abundant bacterial taxa had a stronger potential for nitrogen, sulfur, and methane metabolism than the rare bacterial taxa (Figure 6B), which better supported the views above. In any case, the rarest bacteria occur in fewer grassland soils and were unevenly distributed. This may be attributed to the lower competitive potential and growth rate of rare bacterial taxa, thus limiting their niche width [14]. This study showed that the niche width of abundant bacterial taxa was significantly higher than that of rare bacterial taxa (Appendix A), which proves the abovementioned view.

The high functional redundancy of rare bacterial taxa was to enhance the ability of the system to cope with complex environments and resist environmental disturbance. Even though the rare bacterial taxa had a high degree of functional redundancy, the rare bacterial taxa were more sensitive to grassland degradation. This indicates that the composition of rare species is highly responsive to grassland degradation. Furthermore, the taxonomic and phylogenetic similarities of abundant bacterial taxa were significantly higher than those of rare ones. Some studies suggested that the more susceptible the microbial community is to environmental variation, the more responsive the composition variation for the microbial community. This may indicate that rare bacterial taxa in different degraded grassland soils may be more susceptible to environmental variation than abundant bacterial taxa. This also suggested that the variation of soil environmental factors induced by grassland degradation may be responsible for the different distribution patterns of soil abundant and rare bacterial taxa on the Qinghai-Tibet Plateau.

Assessing the relative ecological contribution of stochastic and deterministic processes to the assembly of the microbial community is an important topic in microbial ecology [24,33]. This study found that the assembly of abundant and rare bacterial taxa in degraded grassland soils on the Qinghai-Tibet Plateau was dominated by stochastic processes. Stochastic processes include unpredictable perturbations, probabilistic dispersion, and random birth-death events [34]. This also implies that grassland degradation caused by grazing makes the ecological assembly processes of soil bacterial communities in the grassland more random, and this stochastic assembly process has a greater impact on abundant bacterial groups than rare bacterial taxa. Deterministic processes are related to ecological selection, in which abiotic and biological factors determine the presence or absence of species and their relative abundance [35]. The assembly processes of variable selection and homogeneous selection belong to deterministic processes to have a higher influence on rare bacterial taxa than abundant bacterial taxa in degraded grassland soils from the Tibetan Plateau.

The combined effects of abiotic environmental filtration and biological interactions on the bacterial community composition are well established. Cooperation between rare and non-rare bacterial taxa may contribute to the resilience of bacterial communities in changing environments [24,36]. However, the competitive potential within the rare bacterial taxa in degraded grassland soils from the Tibetan Plateau was lower than that of the competitive potential between rare and non-rare bacterial taxa (Figure 5A). This may be related to the strong potential of the nutrient metabolism of abundant bacterial taxa in degraded grassland soil. Soil environmental factors such as soil organic matter and pH value are the key factors affecting the assembly of the soil microbial community [10,13]. This study showed that the taxonomic and phylogenetic composition of abundant bacterial taxa in degraded grassland soil were affected by pH, while only the phylogenetic composition of rare bacterial taxa was affected by pH (Appendix A). The phylogenetic composition of rare bacterial taxa rather than abundant ones was also affected by SWC, TN, TOC, TP, and HN. This suggests that rare bacterial taxa were more susceptible to soil nutrient variation than abundant bacterial taxa at the phylogenetic level. This may be because variations in soil nutrients may push bacterial communities toward phylogenetic classifications [24,37]. 

Some potential limitations warrant further discussion. Our results highlight the importance of small-scale grassland environmental changes to the composition and assembly processes of abundant and rare bacterial taxa. However, there is a wide variety of grassland types throughout the Tibetan Plateau. Therefore, more studies on the entire Qinghai-Tibet Plateau need to be carried out in the future combined with remote sensing data sets.

## 5. Conclusions

Grazing will lead to different grassland degradation of the same grassland types under the same climatic environmental region. In this study, we highlight the importance of small-scale grassland environmental changes for the composition and assembly processes of abundant and rare bacterial taxa. The taxonomic composition and phylogenetic composition of rare bacterial taxa were not only affected by grassland vegetation coverage, but also the assembly processes dominated by deterministic processes. This indicated that the composition and assembly of rare bacterial taxa were susceptible to grassland environmental changes caused by grassland degradation. In any case, the distribution of rare bacterial taxa in the different degraded grassland soil was more local than that of abundant bacterial taxa. Therefore, rare bacterial taxa could be considered as ecological indicators of grassland degradation.

## Figures and Tables

**Figure 1 microorganisms-11-00754-f001:**
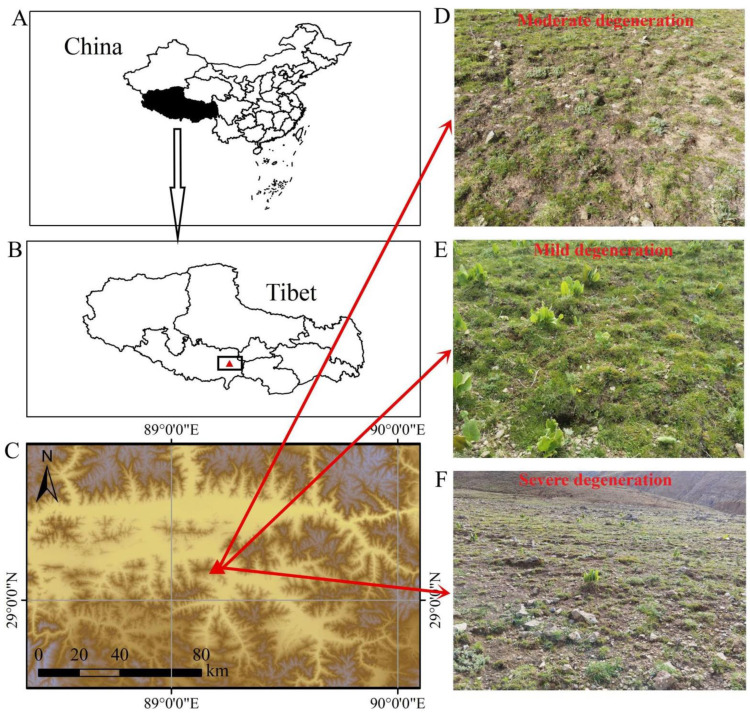
Information of sampling sites in the typical mountain meadow of Qinghai-Tibet Plateau. (**A**–**C**) Geographic information of sampling site. (**D**–**F**) The degree of grassland degradation in the sample site was different.

**Figure 2 microorganisms-11-00754-f002:**
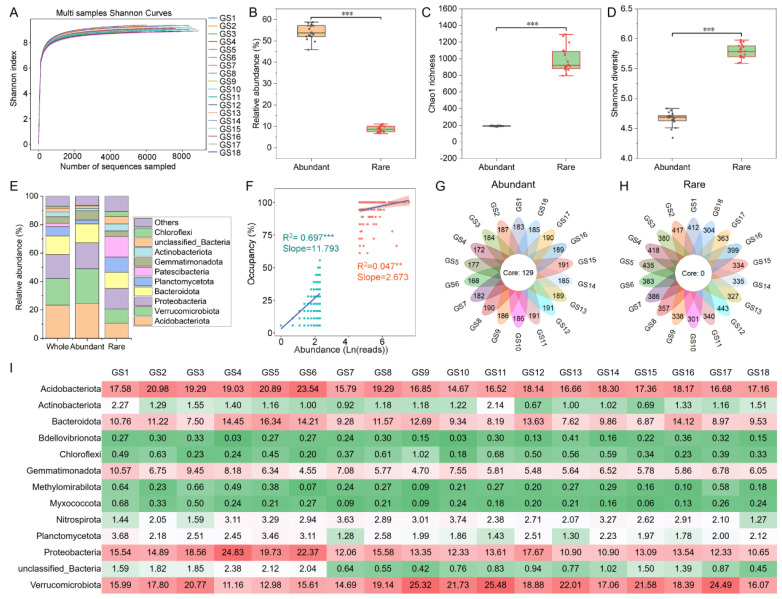
Alpha diversity and composition of abundant and rare bacterial taxa in degraded grassland soils of Tibet. (**A**) Shannon index curve of sequence number showed all soil samples. (**B**) Relative abundance of abundant and rare bacterial taxa in whole bacterial communities. (**C**,**D**) Chao1 richness and Shannon diversity of abundant and rare bacterial taxa. (**E**) Composition of abundant and rare bacterial taxa at the phylum level. (**F**) Abundance-occupancy relationship of abundant and rare bacterial taxa. (**G**,**H**) Petal diagram showing the shared pattern of abundant and rare bacterial taxa in degraded grassland soils. The number on each petal represents the number of ASVs per soil sample, and the number in the middle core represents the number of ASVs shared. (**I**) Heat maps show the relative abundances of the shared abundant bacterial taxa in different degraded grassland soils at the phylum level. Red means having a high relative abundance and green means having a low relative abundance. Asterisks denote significance (**: *p*-value < 0.01; ***: *p*-value < 0.001).

**Figure 3 microorganisms-11-00754-f003:**
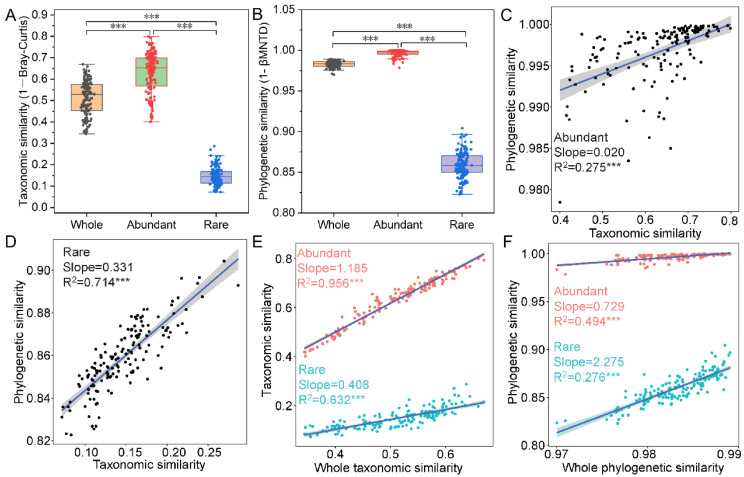
The composition similarity of the bacterial community at the taxonomic and phylogenetic levels. (**A**,**B**) Differential analysis of taxonomic similarity (1–Bray-Curtis) and phylogenetic similarity (1-βMNTD) between abundant and rare bacterial taxa at the ASVs level. (**C**,**D**) Potential association of taxonomic similarity of abundant and rare bacterial taxa with corresponding phylogenetic similarity. (**E**) Potential association of taxonomic similarity of abundant and rare bacterial taxa with the whole bacterial community. (**F**) Potential association of phylogenetic similarity of abundant and rare bacterial taxa with the whole bacterial community. Asterisks denote significance (***: *p*-value < 0.001).

**Figure 4 microorganisms-11-00754-f004:**
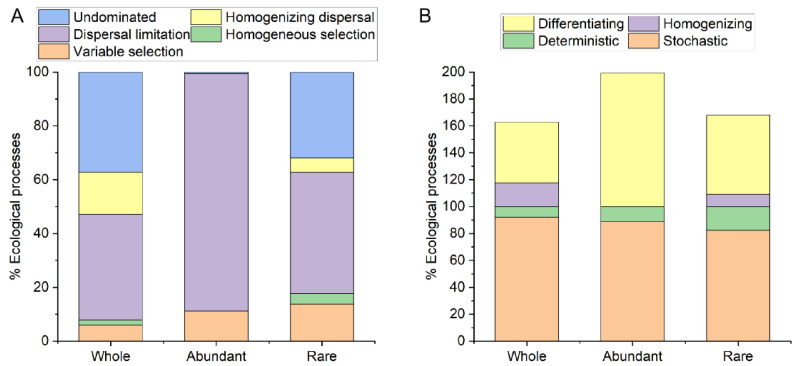
(**A**) Showed that the ecological processes in the assembly of abundant and rare bacterial taxa were calculated via the null model. (**B**) Showed that the Stochastic = Dispersal limitation + Homogenising dispersal + Undominated processes; Deterministic = Variable selection + Homogeneous selection; Homogenising = Homogeneous selection + Homogenising dispersal; Differentiating = Variable selection + Dispersal limitation.

**Figure 5 microorganisms-11-00754-f005:**
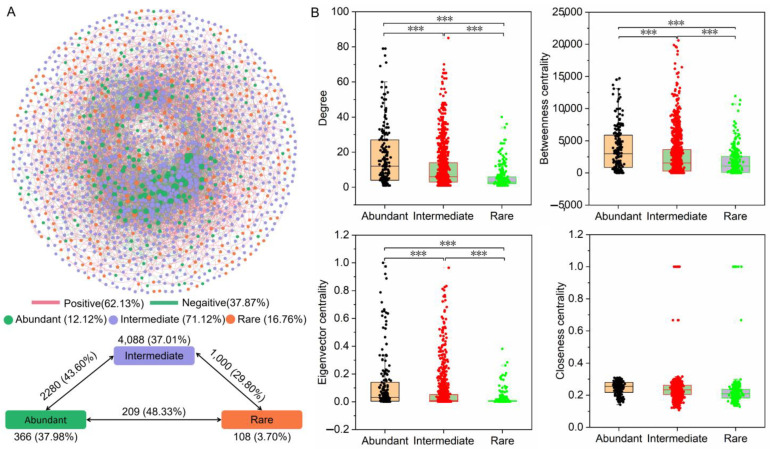
Co-occurrence patterns of the abundant, intermediate, and rare bacterial taxa in degraded grassland soils from Qinghai-Tibet Plateau. (**A**) Network analysis showing the intra-associations and inter-associations between different bacterial taxa. A connection was based on a strong (|r| > 0.7) and significant (*p*-value < 0.01) Spearman’s correlation. The size of each node is proportional to the degree. The numbers outside and inside parentheses represent the total edge numbers and negative edge numbers and their ratio, respectively. (**B**) Comparison of node-level topological features among three different bacterial taxa. Asterisks denote significance (***: *p*-value < 0.001).

**Figure 6 microorganisms-11-00754-f006:**
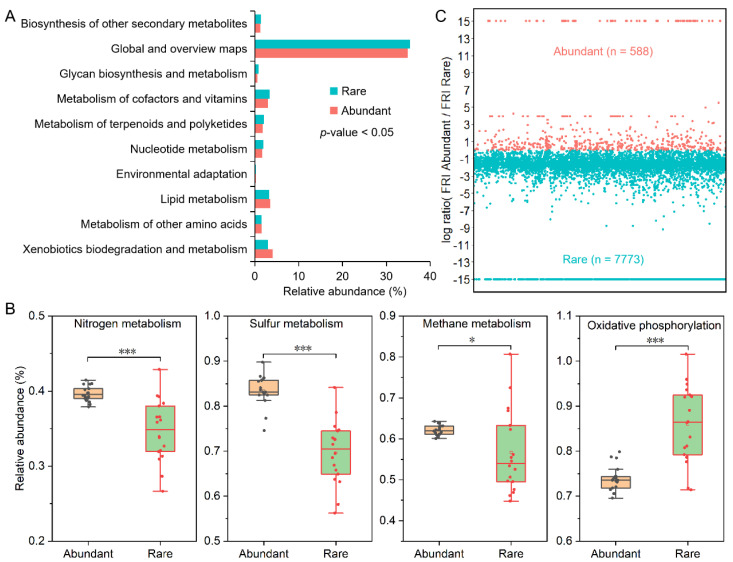
Potential ecological function difference and community functional redundancy of abundant and rare bacterial taxa in degraded grassland soils. (**A**) Difference of Potential ecological function between abundant and rare bacterial taxa in degraded grassland soils. Global and overview maps include the following metabolic pathways: biosynthesis of secondary metabolites, microbial metabolism in diverse environments, biosynthesis of antibiotics, carbon metabolism, 2-oxocarboxylic acid metabolism, fatty acid metabolism, degradation of aromatic compounds, and biosynthesis of amino acids. (**B**) Differences in potential nutrient metabolism between abundant and rare bacterial taxa in degraded grassland soils. Asterisks denote significance (*: *p*-value < 0.05, ***: *p*-value < 0.001). (**C**) Potential functional redundancy of abundant and rare bacterial taxa in degraded grassland soils.

**Table 1 microorganisms-11-00754-t001:** Relationships of taxonomic similarity and phylogenetic similarity of bacterial communities with vegetation characteristics and soil physicochemistry and nutrients were revealed based on Pearson correlation analysis. Asterisks denote significance (*: *p*-value < 0.05; **: *p*-value < 0.01). SWC = soil water content. TN = total nitrogen, TOC = total organic carbon, TP = total phosphorus, HN = hydrolyzable nitrogen, and AP = available phosphorus.

	Taxonomic Similarity			Phylogenetic Similarity		
	Whole	Abundant	Rare	Whole	Abundant	Rare
Vegetation coverage	−0.198 *	−0.150	−0.229 **	−0.245 **	−0.078	−0.268 **
pH	−0.385 **	−0.436 **	−0.15	−0.358 **	−0.366 **	−0.321 **
SWC	−0.370 **	−0.390 **	−0.165 *	−0.314 **	−0.141	−0.279 **
TN	−0.520 **	−0.457 **	−0.517 **	−0.518 **	−0.151	−0.515 **
TOC	−0.475 **	−0.431 **	−0.443 **	−0.465 **	−0.108	−0.450 **
TP	−0.411 **	−0.450 **	−0.279 **	−0.250 **	−0.028	−0.250 **
HN	−0.294 **	−0.214 **	−0.363 **	−0.347 **	0.010	−0.339 **
AP	−0.187 *	−0.127	−0.172 *	−0.229 **	−0.106	−0.131

## Data Availability

The raw sequencing data were deposited in the National Center for Biotechnology Information (NCBI, https://www.ncbi.nlm.nih.gov/, accessed on 21 December 2022) Short Read Archive (SRA) database under accession number PRJNA914547.

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
