# Peer review of "Rare Bacteria Can Be Used as Ecological Indicators of Grassland Degradation"

_microorganisms, 2023, doi:10.3390/microorganisms11030754_

Round 1

Reviewer 1 Report

Manuscript ID: microorganisms-2253643

Review Report

This is a report for the manuscript entitled Rare bacteria can be used as ecological indicators of grassland degradation”.

This manuscript reports an interesting assessment of the bacterial taxa found in soil samples collected in grasslands in the Qinghai-Tibet Plateau, after grazing Tibetan sheep. The identifications were based on the sequencing of the 16S rRNA full-length gene and allowed the presentation of the “diversity and composition of abundant and rare bacterial taxa” (section 3.1). Despite the clarity of these results, they should be more comprehensive in what concerns the identifications, only presented at the phylum level and not made publicly available; it is a relevant matter, to provide that information in a public database.

Moreover, although the number of ASVs is indicated in Fig 2 G-H, and “taxonomic annotation of feature sequences was processed by bayesian classifier and blast using Silva 138 as a reference database”, that information was not provided, neither was the lowest level of the identifications (at species level? at genus level?). The composition similarity of bacterial communities at the taxonomic and phylogenetic levels were addressed (section 3.2) but, once more, no information was given about the level/s achieved in the identifications. This is also relevant, mainly whenever to put in practice what is suggested in the manuscript. This subject should be addressed.

Besides the concerns previously provided, the methodological approaches applied were adequate and rendered data to comprehensively and adequately address the subjects of the following sections: Responses of bacterial communities to vegetation, soil physicochemical characteristics, and nutrients (3.2) (suggested title), “Assembly processes of abundant and rare bacterial taxa in degraded grassland soil” (3.3), “Co-occurrence network of abundant and rare bacterial taxa in degraded grassland soil” (3.4), and “Potential ecological function of abundant and rare bacterial taxa in degraded grassland soils” (3.5).

The manuscript largely evidences, along the different sections, the complex responses of the two groups (abundant and rare) of bacterial taxa to environmental changes under grassland degradation, also concluding that “the distribution of rare bacterial taxa in the different degraded grassland soil was more local than that of abundant bacterial taxa”, which would allow considering rare bacteria as ecological indicators of grassland degradation.

Although the manuscript includes clear and carefully made tables, figures, and supplementary file, it may still be improved. The comments and suggestions in the appended PDF version (microorganisms-2253643-peer-review-v1 (REVIEW), are intended to be a contribution for this purpose.

Author Response

ANS: We appreciate the reviewer’s positive evaluation of our work. Our deepest gratitude goes to you for your careful work and thoughtful suggestions that have helped improve this paper substantially. According to the comments and suggestions in the appended PDF version, we have revised the manuscript accordingly, and these revision was highlighted in blue color.

We provied the information of public database In the Data Availability Statement parts: The raw sequencing data were deposited in the National Center for Biotechnology Information (NCBI, https://www.ncbi.nlm.nih.gov/) Short Read Archive (SRA) database under accession number PRJNA914547.

The composition similarity of bacterial community at the taxonomic and phylogenetic level was estimated based on the ASVs level via the “vegan” package and the “picante” package, respectively. (Line 126). Besides, this study focused on the response of taxonomic composition and phylogeny of rich and rare bacterial groups to environmental changes in degraded grassland at the ASV level, as well as the response of community aggregation mechanisms at the ASVs level to environmental changes in degraded grassland. So the composition of the species is not described at a higher taxonomic level.

Reviewer 2 Report

The present research investigated the variation of rare and abundant bacterial taxa in grazing grassland in Tibetan, which is an important area in ecological safety. The study was well designed and the manuscript was written generally good. I have three major concerns:

1. Rare bacterial taxa are more sensitive to grassland degradation, are they functionally important in the grassland? As the bacterial community have the functional redundancy, the disappearance of the rare bacterial taxa might not be of that significance.

2. The impact of grazing gradients was not shown.

3. The definition of “abundant taxa” and “rare taxa” should be clearly stated in the method section.

Some minor points as below.

Abstract:

Line 26: the “aggregation mechanism” should be replaced by “assembly mechanism”.

Figure 1, A map of China with the location of Tibet should be supplemented.

Line 129-130: “The diversity……full-length gene sequencing.” The sentence can be removed as this is a part of methods.

Author Response

ANS: Thank you very much for providing us with this opportunity to revise the manuscript. Our deepest gratitude goes to you for your careful work and thoughtful suggestions that have helped improve this paper substantially. We apologize for the language problems in the original manuscript. The language presentation was improved with assistance from a native English speaker with appropriate research background. These revision was highlighted in green color.

Comment <1>. Rare bacterial taxa are more sensitive to grassland degradation, are they functionally important in the grassland? As the bacterial community have the functional redundancy, the disappearance of the rare bacterial taxa might not be of that significance.

ANS: Our deepest gratitude goes to you for your careful work and thoughtful suggestions. In this study, rare bacterial taxa have strong oxidative phosphorylation and other potential ecological functions. The high functional redundancy of rare bacterial taxa is to enhance the ability of the system to cope with complex environment and resist environmental disturbance. Even though the rare bacterial taxa had a high degree of functional redundancy, while the rare bacterial taxa were more sensitive to grassland degradation. This indicates that the composition of rare species is highly responsive to grassland degradation. (Line 328-332)

Comment <2>. The impact of grazing gradients was not shown.

ANS: Thank you for the suggestion. This study was conducted on a grassland after grazing by local herders. As Tibetan sheep like to gather together to forage, they also have different preferences for different forage types. This kind of trampling caused by preferential feeding process led to the different degradation of grassland. Therefore, we set out to investigate the response of this abundant and rare bacterial taxa to this patchy grassland degradation. Thus, we didn't have statistics on grazing intensity at that time.

Comment <3>. The definition of “abundant taxa” and “rare taxa” should be clearly stated in the method section.

ANS: Thank you for the suggestion. Combined with Reviewer 1's suggestions, we have added the information required as explained above. According to previous studies [29, 37], we defined ASVs at the regional level with average relative abundances >0.10% as “abundant,” thosewith average relative abundances <0.01% as “rare” and those inbetween as “intermediate”. (Line 122-124)

Comment <4>. Line 26: the “aggregation mechanism” should be replaced by “assembly mechanism”.

ANS: Thank you for your suggestion. The “aggregation mechanism” was replaced by “assembly mechanism”. (Line 26)

Comment <5>. Figure 1, A map of China with the location of Tibet should be supplemented.

ANS: Thank you for your suggestion. We have added A map of China with the location of Tibet. (Line 95)

Comment <6>. Line 129-130: “The diversity……full-length gene sequencing.” The sentence can be removed as this is a part of methods.

ANS: Thank you for your suggestion. We deleted the sentence. (Line 137)

Round 2

Reviewer 1 Report

The revised version of the manuscript answered the major issues presented in my first revision, except in what concerns the two sentences in lines 43-47, which still do not make sense and should be revised again.

Author Response

ANS: Our deepest gratitude goes to you for your careful work and thoughtful suggestions that have helped improve this paper substantially. Based on your comments and those of academic reviewers, we have revised this content and deleted the meaningless expressions in lines 43-47. Besides, we rechecked the entire text, and some other language changes were highlighted in green.
